# Semantic Perturbations with Normalizing Flows for Improved Generalization

**Oğuz Kaan Yüksel** [1]  **Sebastian U. Stich** [1]  **Martin Jaggi** [1]  **Tatjana Chavdarova** [1]

## Abstract

Several methods from two separate lines of works, namely, data augmentation (DA) and adversarial training techniques, rely on perturbations done in *latent* space. Often, these methods are either non-interpretable due to their non-invertibility or are notoriously difficult to train due to their numerous hyperparameters. We exploit the exactly reversible encoder-decoder structure of normalizing flows to perform perturbations in the latent space. We demonstrate that these on-manifold perturbations match the performance of advanced DA techniques—reaching $96.6\%$ test accuracy for CIFAR-10 using ResNet-18 and outperform existing methods particularly in low data regimes—yielding 10–25% relative improvement of test accuracy from classical training. We find our latent *adversarial* perturbations, adaptive to the classifier throughout its training, are most effective.

## 1. Introduction

Successfully applying Deep Neural Networks (DNNs) in real world tasks in large part depends on the availability of large annotated datasets for the task at hand. Thus, besides several overfitting techniques, *data-augmentation* (DA) often remains a "mandatory" step when deploying DNNs in practice. Traditional DA techniques consist of applying a predefined set of transformations to the training samples that do not change the class label. As this approach is limited to making the classifier robust *solely* to the fixed set of hard-coded transformations, advanced methods incorporate more loosely defined transformations in the data space (Zhang et al., 2018a; DeVries & Taylor, 2017; Yun et al., 2019). Furthermore, recently proposed DA techniques exploit the *latent space* to perform such transformations (Antoniou et al., 2017; Zhao et al., 2018), while typically solving the model's non-invertability by training a *separate* model (Zhao et al.,

[1]EPFL, Lausanne, Switzerland. Correspondence to: Oğuz Kaan Yüksel <oguz.yuksel@epfl.ch>.

Third workshop on *Invertible Neural Networks, Normalizing Flows, and Explicit Likelihood Models* (ICML 2021). Copyright 2021 by the author(s).

2018), thus making them hard to train.

A separate line of work focuses on *adversarial training* (see (Biggio & Roli, 2018) and references therein), where the final model is trained with samples perturbed in a way that makes its prediction incorrect, called *adversarial* samples. However, further empirical studies showed that such training reduces the "clean" samples accuracy, indicating the two objectives are competing (Tsipras et al., 2019b; Su et al., 2018). Stutz et al. (2019) postulate that this robustness-generalization trade-off appears due to using off-manifold adversarial attacks that leave the data-manifold, and that 'on-manifold adversarial attacks' can improve generalization. Thus, the authors proposed to use perturbations in the latent space of a generative model, VAE-GAN (Larsen et al., 2016; Rosca et al., 2017). However, as this method relies on the VAE-GAN model which is particularly hard to train—since in addition to GAN training it involves hard to tune hyperparamaters balancing the VAE and GAN objectives—its usage remained limited.

Motivated by the advantages of *normalizing flows* (NF) relevant to these two lines of works, we employ NFs (*e.g. Glow,* Kingma & Dhariwal, 2018), to define entirely unsupervised augmentations—contrasting with pre-defined fixed transformations—with the same goal of improving the generalization of deep classifiers. In particular, NF models offer appealing advantages for latent space perturbations, such as: (i) exact latent-variable inference and log-likelihood evaluation, and (ii) efficient inference and synthesis that can be parallelized (Kingma & Dhariwal, 2018).

**Related works.** Several works learn useful DA policies, for instance by optimization (Fawzi et al., 2016; Ratner et al., 2017), Reinforcement Learning techniques (Cubuk et al., 2019; 2020; Zhang et al., 2020b), specifically trained generator networks (Peng et al., 2018) or assisted by generative adversarial networks (Perez & Wang, 2017; Antoniou et al., 2017; Zhang et al., 2018b; Tran et al., 2020). Several methods traverse the latent space to find virtual data samples that are missclassified (Baluja & Fischer, 2017; Song et al., 2018; Xiao et al., 2018; Zhang et al., 2020a). Complementary, the connection of adversarial learning and generalization has also been studied in (Tanay & Griffin, 2016; Rozsa et al., 2016; Jalal et al., 2017; Tsipras et al., 2019a; Gilmer et al., 2018; Zhao et al., 2018).

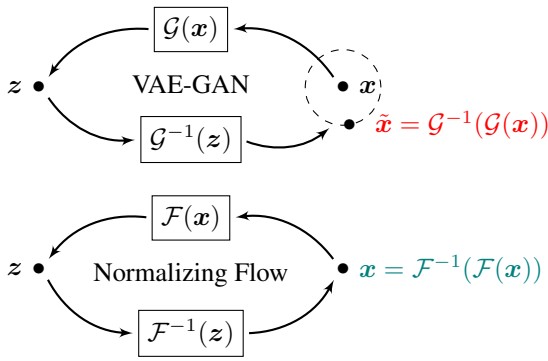

**Figure 1. Exactness of NF encoding-decoding**. Here $\mathcal{F}$ denotes the bijective NF model, and $\mathcal{G}/\mathcal{G}^{-1}$ encoder/decoder pair of inexact methods such as VAE or VAE-GAN which, due to inherent decoder noise, is only approximately bijective.

**Overview of contributions.** We exploit the *exactly reversible* encoder-decoder structure of NFs to perform efficient and controllable augmentations in the learned on-manifold space. We demonstrate that our on-manifold perturbations consistently outperform the standard training on CIFAR-10/100 using ResNet-18. Moreover, in a low-data regime, such training yields up to $25\%$ relative improvement from classical training, of which—as most effective—we find the adversarial perturbations that are adaptive to the classifier throughout its training, see §3.

## 2. Perturbations in Latent Space

The invertibility of normalizing flows enables bidirectional transitions between image and latent space, see Figure 1. This in turn allows for applying perturbations directly in the latent space rather than image space. We denote a trained NF model, mapping data manifold $\mathcal{X}$ to latent space $\mathcal{Z}$ as, $\mathcal{F} : \mathcal{X} \to \mathcal{Z}$. Given a perturbation function $\mathcal{P} : \mathcal{Z} \to \mathcal{Z}$, defined over the latent space, we define its counterpart in image space as $\mathcal{F}^{-1}(\mathcal{P}(\mathcal{F}(\boldsymbol{x})))$.

Our goal is to define latent perturbation function $\mathcal{P}(\cdot)$ such that we obtain identity-preserving semantic modifications over the original image $\boldsymbol{x}$ in the image domain. To this end, we limit the structure of possible $\mathcal{P}$ in two-ways. Firstly, we directly consider incremental perturbations of the form $\boldsymbol{z} + \mathcal{P}(\boldsymbol{z})$. Secondly, we use an $\epsilon$ parameter to control the size of perturbation allowed. More precisely, we have:

$$\mathcal{F}^{-1}\big(\mathcal{F}(\boldsymbol{x}) + \mathcal{P}(\mathcal{F}(\boldsymbol{x}), \epsilon)\big).$$

For brevity, we refer to $\mathcal{P}$ as "latent attacks" (LA) and we consider two variants, described below.

### 2.1. Randomized Latent Attacks

At training time, given a datapoint $\boldsymbol{x}_i$, with $1 \leq i \leq n$, using trained normalizing flow we obtain its corresponding

latent code $\boldsymbol{z}_i = \mathcal{F}(\boldsymbol{x}_i)$.

Primarily, as perturbation function we consider a simplistic Gaussian noise in the latent space:

$$\mathcal{P}_{rand}(\cdot, \epsilon) = \epsilon \cdot \mathcal{N}(0, \mathbf{I}), \qquad \textbf{(Randomized–LA)}$$

which is independent from $\boldsymbol{z}_i$. Any such distribution around the original $\boldsymbol{z}_i$ is equivalent to sampling from the learned manifold. In this case, the normalizing flow *pushes forward* this simple Gaussian distribution centered around $\boldsymbol{z}_i$ to a distribution on the image space around $\boldsymbol{x}_i = \mathcal{F}^{-1}(\boldsymbol{z}_i)$. Thus, sampling from the simple prior distribution $\mathcal{N}(0, \mathbf{I})$ is equivalent to sampling from a complex conditional distribution around the original image over the data manifold.

We also define norm truncated versions as follows:

$$\mathcal{P}_{rand}^{\ell_p}(\cdot, \epsilon) = \Pi(\epsilon \cdot \mathcal{N}(0, \mathbf{I})),$$

where $\ell_p$ denotes the selected norm, e.g. $\ell_2$ or $\ell_\infty$. For $\ell_2$ norm, $\Pi$ is defined as $\ell_2$ norm scaling, and for $\ell_\infty$, $\Pi$ is the component-wise clipping operation defined below:

$$(\Pi(\boldsymbol{x}))_i := \max(-\epsilon, \min(+\epsilon, \boldsymbol{x}_i)).$$

### 2.2. Adversarial Latent Attacks

Analogous to the above randomized latent attacks, at train time, given a datapoint $\boldsymbol{x}_i$ and it's associated label $l_i$, with $1 \leq i \leq n$, using trained normalizing flow we obtain its corresponding latent code $\boldsymbol{z}_i = \mathcal{F}(\boldsymbol{x}_i)$.

We search for $\Delta_{\boldsymbol{z}_i} \in \mathcal{Z}$ such that the loss obtained of the generated image $\tilde{\boldsymbol{x}}_i = \mathcal{F}^{-1}(\boldsymbol{z}_i + \Delta_{\boldsymbol{z}_i})$ is maximal:

$$\Delta_{\boldsymbol{z}_i}^\star = \underset{\|\Delta_{\boldsymbol{z}_i}\|_{l_p} \leq \epsilon}{\arg\max} \; \mathcal{L}_\theta(\mathcal{F}^{-1}(\boldsymbol{z}_i + \Delta_{\boldsymbol{z}_i}), l_i),$$

$$\mathcal{P}_{adv}^{\ell_p}(\boldsymbol{z}_i, \epsilon) = \Delta_{\boldsymbol{z}_i}^\star, \qquad \textbf{(Adversarial–LA)}$$

where $\mathcal{L}_\theta$ is the loss function of the classifier, and $\ell_p$ denotes the selected norm, *e.g.* $\ell_2$ or $\ell_\infty$.

In practice, we define the number of steps $k$ and the step size $\alpha$ to optimize for $\Delta_{\boldsymbol{z}_i}^\star \in \mathcal{Z}$ (as in Stutz et al. (2019); Wong & Kolter (2021)), and we have the following procedure:

- Initialize a random $\Delta_{\boldsymbol{z}_i}^0$ with $\|\Delta_{\boldsymbol{z}_i}^0\|_{\ell_p} \leq \epsilon$.
- Iteratively update $\Delta_{\boldsymbol{z}_i}^j$ for $j = 1, \dots, k$ number of steps with step size $\alpha$ as follows:

$$\Delta_{\boldsymbol{z}_i}^j = \Pi\Big(\Delta_{\boldsymbol{z}_i}^{j-1} + \alpha \cdot \frac{\nabla \mathcal{L}_\theta(\mathcal{F}^{-1}(\boldsymbol{z}_i + \Delta_{\boldsymbol{z}_i}^{j-1}), l_i)}{\|\nabla \mathcal{L}_\theta(\mathcal{F}^{-1}(\boldsymbol{z}_i + \Delta_{\boldsymbol{z}_i}^{j-1}), l_i)\|_{\ell_p}}\Big)$$

where $\Pi$ is the projection operator that ensures $\|\Delta_{\boldsymbol{z}_i}^j\|_{\ell_p} \leq \epsilon$ and the gradient is with respect to $\Delta_{\boldsymbol{z}_i}^j$.
- Output $\mathcal{P}_{adv}(\boldsymbol{z}_i, \epsilon) = \Delta_{\boldsymbol{z}_i}^k$

Table 1. *Test accuracy* (%) on **CIFAR-10**, in the *low-data regime* compared to the *full train set*. For the former, we train generative models and classifiers on the same 5% of the training set and evaluate classifiers on the full test set. In addition to standard training, we consider training with commonly used data augmentations (DA) in the image space, which includes rotation and horizontal flips (Zagoruyko & Komodakis, 2016), as well as more recent *Cutout* (DeVries & Taylor, 2017) and *Mixup* (Zhang et al., 2018a) methods. See §3.1 for a discussion.

| Method | Low-data | Full-set |
|---|---|---|
| Standard (no DA) | 49.8 | 89.7 |
| Standard + common DA | 64.1 | 95.2 |
| VAE-GAN (Stutz et al., 2019) | 58.9 | 94.2 |
| Cutout (DeVries & Taylor, 2017) | 66.8 | 96.0 |
| Mixup (Zhang et al., 2018a) | 73.4 | 95.9 |
| Randomized–LA (ours) | 70.1 | 96.3 |
| Adversarial–LA (ours) | **80.4** | **96.6** |

For the case of $\ell_\infty$, we replace normalization of gradient with $sign(\cdot)$ operator, i.e.:

$$\Delta_{\boldsymbol{z}_i}^j = \Pi\Big(\Delta_{\boldsymbol{z}_i}^{j-1} + \alpha \cdot sign\big(\nabla\mathcal{L}_\theta(\mathcal{F}^{-1}(\boldsymbol{z}_i + \Delta_{\boldsymbol{z}_i}^{j-1}), l_i)\big)\Big)$$

and use component-wise clipping for projection, equivalent to the standard $\ell_\infty$ PGD adversarial attack of Madry et al..

Similarly, as NFs directly models the underlying data manifold, this perturbation is equivalent to a search over the *on-manifold* adversarial samples (Stutz et al., 2019).

## 3. Experiments

### 3.1. Generalization on CIFAR-10

We are primarily interested in the performance of our perturbations in the low-data regime, when using only a small subset of CIFAR-10 as the training set. We train ResNet-18 classifiers on only 5% percent of the full training set and evaluate models on the full test set.

We compare our methods with some of the most commonly used data augmentations methods such as *Cutout* (DeVries & Taylor, 2017) and *Mixup* (Zhang et al., 2018a), as well as with the VAE-GAN based approach (Stutz et al., 2019). For (Stutz et al., 2019), we use the authors' implementation.

For (DeVries & Taylor, 2017), we report the best test accuracy observed among a grid search on the learning rate $\eta \in \{0.1, 0.01\}$. Similarly, for (Zhang et al., 2018a), we report the best accuracy among grid search on learning rate $\eta \in \{0.1, 0.01\}$ and mixup coefficient $\lambda \in \{.1, .2, .3, .4, 1.0\}$. For Randomized–LA, we use $\ell = \ell_\infty, \epsilon = 0.25$, and for Adversarial–LA, we use $\ell = 2, \epsilon = 1.0, \alpha = 0.5, k = 3$.

Table 1 summarizes our generalization experiments in the

Table 2. **Cross-datasets experiments.** *Test accuracy* (%) on **CIFAR-100**, in the *low-data regime*, where we use 10% of the training set and the full test set. The normalizing flow used to generate training samples is trained on **CIFAR-10**.

| Method | Test | Improvement |
|---|---|---|
| Standard | 36.4 | – |
| Randomized–LA, $\ell=\ell_\infty, \epsilon=.2$ | 39.7 | +3.3 |
| Randomized–LA, $\ell=\ell_\infty, \epsilon=.3$ | 41.0 | +4.6 |
| Randomized–LA, $\ell=\ell_2, \epsilon=10$ | 40.4 | +4.0 |
| Randomized–LA, $\ell=\ell_2, \epsilon=20$ | **42.3** | +5.9 |
| Adversarial–LA, $\ell=\ell_2, \alpha=.5, k=3$ | **45.0** | +8.6 |

low data regime—using only 5% of CIFAR-10 for training, compared to the full CIFAR-10 training set. Both Randomized–LA and Adversarial–LA notably outperform the standard training baseline. In particular, we observe that (i) our simplistic Randomized–LA method already outperforms some recent strong data augmentation methods, and (ii) Adversarial–LA achieves *best* test accuracy for both low-data and full-set regimes. See §3.3 below for additional benchmarks with VAE-GAN (Stutz et al., 2019) and §B.3 for additional results with different attack parameters.

### 3.2. Cross-Dataset Experiments

To further analyze potential applications of our NF based latent attacks on real-world use cases, we conduct the following experiment. Assuming we have available a relevant large-scale dataset, a question arises if the NF within our approach can be pre-trained on it, and used for training the classifier on a different dataset. In particular, we use CIFAR-10 to train the NF model, and then our latent attacks to train a classifier on 10% of the CIFAR-100 dataset.

Table 2 shows our results for a selection of latent attacks. Randomized–LA and Adversarial–LA achieve 16% and 24% percent improvements over the standard baseline, respectively. The results indicate that NFs are capable of transferring useful augmentations across datasets. See §B.2 for an additional results with SVHN and CIFAR-10.

### 3.3. Additional Comparison with VAE-GAN

We study the performance of our latent perturbation-based training strategies in varying settings, starting from low-data regime to full-set. We reproduce the classifier and the hyperparameter setup used in (Stutz et al., 2019), and use analogous setup for our method. For the reported VAE-GAN results, we used the source code provided by the authors[1]. For our Randomized–LA, we use perturbations of size $\epsilon=0.15$ and for Adversarial–LA, we use $\epsilon=0.05, \alpha=0.01$ and number of steps $k = 10$.

[1] https://github.com/davidstutz/disentangling-robustness-generalization

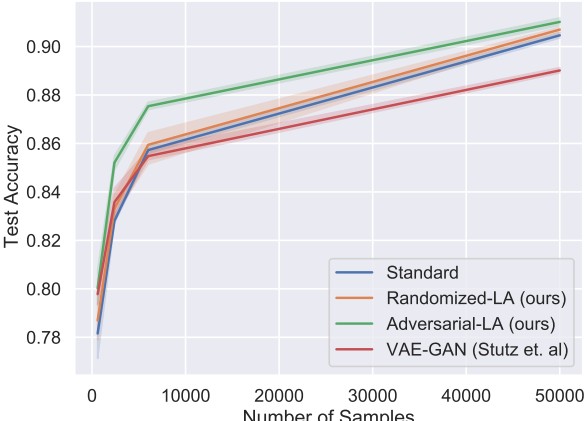

*Figure 2.* Test accuracy (y-axis)—on full test set, for a varying number of *training* samples (x-axis), on **FashionMNIST**. To replicate the setup of VAE-GAN (Stutz et al., 2019), only a portion of the dataset (x-axis) is used to train the classifier, while the corresponding generative model is trained on the full dataset. We run each experiment with three different random seeds, and report the mean and standard deviation of the test accuracy. See §3.3.

Figure 2 shows our average results for 3 runs with training sizes in $\{600, 2400, 6000, 50000\}$. We observe that Randomized–LA performs comparatively to the standard training baseline, whereas Adversarial–LA outperforms the standard baseline across all train set sizes. Note that the difference to the standard baselines shrinks as we increase the number of samples available to the classifiers.

Inline with our results, Stutz et al. (2019) report diminishing performance gains for increasingly challenging datasets such as FashionMNIST to CelebA, when using therein VAE-GAN based approach. One potential cause could be the *approximate* encoding and decoding mappings, and/or sensitivity to hyper-parameter tuning. Indeed, our results support the numerous appealing advantages of NF models for latent space perturbations, and indicate that they have better capacity to produce useful augmented training samples.

## 4. Discussion

**Exact Coding.** As noted in §2, normalizing flows can perform exact encoding and decoding by their construction. That is, the decoding operation is exactly the reverse of the encoding operation. Any continuous encoder maps a neighborhood of a sample to some neighborhood of its latent representation. However, the invertibility of normalizing flows also maps *any neighborhood of latent code to a neighborhood of the original sample*. This property has significant advantages over any other approximate invertible encoder-decoder methods including VAE-GANs, for defining perturbations in latent space.

**Increasing Effective Dataset Size.** The primary advantage of exact coding is that the generated samples via latent

perturbations improve the generalization performance of classifiers, as we show in §3.1. To understand why this occurs, consider the limit case $\epsilon \to 0$ for a latent perturbation. For a numerically stable NF, this implies that we recover the original data samples, hence the original data manifold. As we increase the $\epsilon$, we "enlarge" our manifold simultaneously from all data samples. Thus, by increasing $\epsilon$, we add further plausible data points to our training set as long the learned latent representation is a good approximation of the underlying data manifold. This does not necessarily hold for approximate mappings due to inherent *decoder noise*.

**Controllability.** In §2, we introduced two variants of latent perturbations with normalizing flows. These two variants define different local objectives around the latent code of the original sample. The Randomized–LA defines a sampling operation on the data manifold, whereas the Adversarial–LA defines a stochastic search procedure to find samples attaining high classifier losses. Here, we exploit the diffeomorphism provided by normalizing flow to efficiently map a complex sampling operation—sampling from data manifold, or a complex search operation—finding on-manifold adversarial samples, to the latent space. Combined with simple prior structure of normalizing flows, this allow for future possibilities on designing efficient algorithms tackling various on manifold problems §5.

**Compatibility with Data Augmentations.** It is important to note that our method is orthogonal to image space data augmentation methods. In other words, we can train normalizing flows with commonly used data augmentations. Indeed in our experiments, we observe that trained models apply some of the training-time augmentations such as cropping. This allows us to encode and decode *augmented* samples as well as original samples of CIFAR-10. Additionally, we can use more advanced methods such as DeVries & Taylor (2017); Zhang et al. (2018a) concurrently with our latent perturbations to train classifiers.

## 5. Conclusion

Motivated by the numerous advantages of normalizing flows, we propose flow-based latent perturbation methods to augment the training datasets, to train a classifier. Our empirical results on several real-world datasets demonstrate the efficacy of these generative models for improved test accuracy both in full and in low-data regimes.

Further directions include exploiting potentially more complex prior structures to design efficient flow-based algorithms tackling on-manifold sampling or optimization problems. For example, using NF models with explicit parametrization of specific semantic transformations (e.g., zoom or orientation) would enable the training of more robustly generalizing classifiers.

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

# A. Details on the implementation

In this section, we list all details on the implementation.

**Source Code.** Our source code is provided in this repository: `https://github.com/okyksl/flow-lp`.

## A.1. Architectures

**Generative model (NF) architecture.** We use *Glow* (Kingma & Dhariwal, 2018) for the normalizing flow architecture. For the MNIST (Lecun et al., 1998) and FashionMNIST (Xiao et al., 2017) experiments, we use a conditional, 12-step, Glow-coupling-based architecture similar to (Ardizzone et al., 2019). See Table 3 for details. For the CIFAR-10/100 (Krizhevsky & Hinton, 2009) and SVHN (Netzer et al., 2011) experiments, we use the original Glow architecture described in (Kingma & Dhariwal, 2018), *i.e.*, 3 scales of 32 steps each containing activation normalization, affine coupling and invertible $1\times1$ convolution. We adapt an existing PyTorch implementation in [2] to better match the original Tensorflow implementation in [3]. For more details on multi-scale architecture in normalizing flows, see (Dinh et al., 2017).

*Table 3.* Normalizing flow architectures used for our experiments on **MNIST** and **FashionMNIST**. With $c_{in} \to y_{out}$ we denote the number of channels of the input and output of the layer. With $\oplus$ we denote concatenation operation. We use the implementation provided in `https://github.com/VLL-HD/FrEIA`. For more details on affine coupling layers, see (Kingma & Dhariwal, 2018).

| **Generative Model** |
| --- |
| *Input:* $\boldsymbol{x} \in \mathbb{R}^{784}, \boldsymbol{y} \in \mathbb{R}^{10}$ |
| GLOWCouplingBlock |
| PermuteRandom |
| GLOWCouplingBlock |
| PermuteRandom |
| ... |
| x9 |
| ... |
| GLOWCouplingBlock |
| PermuteRandom |

| **GLOWCouplingBlock** |
| --- |
| *Input:* $\boldsymbol{x} \in \mathbb{R}^{784}, \boldsymbol{y} \in \mathbb{R}^{10}$ |
| split $\boldsymbol{x} \to \boldsymbol{x}_1, \boldsymbol{x}_2$ ($784 \to 392, 392$) |
| subnet $\boldsymbol{x}_2 \oplus \boldsymbol{y} \to \mathbf{s}_1, \mathbf{t}_1$ ($402 \to 392, 392$) |
| affine coupling $\boldsymbol{x}_1, \mathbf{s}_1, \mathbf{t}_1 \to \boldsymbol{z}_1$ ($3\times392 \to 392$) |
| subnet $\boldsymbol{z}_1 \oplus \boldsymbol{y} \to \mathbf{s}_2, \mathbf{t}_2$ ($402 \to 392, 392$) |
| affine coupling $\boldsymbol{x}_2, \mathbf{s}_2, \mathbf{t}_2 \to \boldsymbol{x}_2'$ ($3\times392 \to 392$) |
| concat. $\boldsymbol{z}_1 \oplus \boldsymbol{z}_2$ ($392, 392 \to 784$) |

| **Subnets** |
| --- |
| *Input:* $\boldsymbol{x} \in \mathbb{R}^{402}$ |
| linear ($402 \to 512$) |
| ReLU |
| linear ($512 \to 784$) |
| split ($784 \to 392, 392$) |

**Classifier architecture.** For our experiments on MNIST, we use LeNet-5 (Lecun et al., 1998) with replaced nonlinearity– instead of $\tanh$ we use $ReLU$, and we initialize the network parameters with truncated normal distribution $\sigma = 0.1$. For the FashionMNIST experiments, we use the same classifier as used in (Stutz et al., 2019). See Table 4 for more details. For CIFAR-10/100 and SVHN, we use the ResNet-18 architecture as implemented in (DeVries & Taylor, 2017; Zhang et al., 2018a). This ResNet-18 includes slight modifications over the standard ResNet-18 architecture in order to achieve better performance on CIFAR-10/100. See [4] and [5] for implementation. In particular, the first layer is changed to a $3 \times 3$ convolution with stride 1 and padding 1, from the original $7 \times 7$ convolution with stride 2 and padding 3. Additionally, the following max-pooling layer is removed.

## A.2. Hyperparameters

**Generative Models.** For MNIST and FashionMNIST, we use the Adam (Kingma & Ba, 2014) optimizer with a batch size of 100 and learning rate of $10^{-6}$ for 100 epochs to train normalizing flows. For CIFAR-10 and SVHN, we use the Adamax (Kingma & Ba, 2014) optimizer with learning rate of 0.0005 and weight decay of 0.00005. We use warmup learning rate scheduling for the first 500.000 steps of the training. That is, the learning rate is linearly increased from 0 to the base learning rate 0.0005 in 500.000 steps.

For VAE-GAN training, we run the implementation provided by authors[6] with the default architectures and parameters. That

---

[2] `https://github.com/chrischute/glow`
[3] `https://github.com/openai/glow`
[4] `https://github.com/facebookresearch/mixup-cifar10`
[5] `https://github.com/uoguelph-mlrg/Cutout`
[6] `https://github.com/davidstutz/disentangling-robustness-generalization`

*Table 4.* Convolutional Neural Network (CNN) architectures used for our experiments on **MNIST** and **FashionMNIST**. We use *ker* and *pad* to denote *kernel* and *padding* for the convolution layers, respectively. With $h \times w$ we denote the kernel size. With $c_{in} \rightarrow y_{out}$ we denote the number of channels of the input and output of the layer.

| **LeNet-5** |
| --- |
| *Input:* $\boldsymbol{x} \rightarrow \mathbb{R}^{1 \times 28 \times 28}$ |
| convolution (ker: $5 \times 5$, $1 \rightarrow 6$; stride: 1; pad:2) |
| ReLU |
| AvgPool2d (ker: $2 \times 2$) |
| convolution (ker: $5 \times 5$, $6 \rightarrow 16$; stride: 1; pad:0) |
| ReLU |
| AvgPool2d (ker: $2 \times 2$) |
| Flatten ($16 \times 5 \times 5 \rightarrow 400$) |
| linear ($400 \rightarrow 120$) |
| ReLU |
| linear ($120 \rightarrow 84$) |
| ReLU |
| linear ($120 \rightarrow 10$) |
| ReLU |

| **CNN from (Stutz et al., 2019)** |
| --- |
| *Input:* $\boldsymbol{x} \in \mathbb{R}^{1 \times 28 \times 28}$ |
| convolution (ker: $4 \times 4$, $1 \rightarrow 16$; stride: 2; pad:1) |
| Batch Normalization |
| ReLU |
| convolution (ker: $4 \times 4$, $16 \rightarrow 32$; stride: 2; pad:1) |
| Batch Normalization |
| ReLU |
| convolution (ker: $4 \times 4$, $32 \rightarrow 64$; stride: 2; pad:1) |
| Batch Normalization |
| ReLU |
| Flatten ($64 \times 3 \times 3 \rightarrow 576$) |
| linear ($576 \rightarrow 100$) |
| linear ($100 \rightarrow 10$) |

is, for FashionMNIST, we use $\beta = 2.75$, $\gamma = 1$, $\eta = 0$ and latent space size of 10. We use Adam optimizer with a batch size of 100, learning rate of 0.005, weight decay of 0.0001 and train VAE-GANs for 60 epochs with an exponential decay scheduling of 0.9 for the learning rate. For CIFAR-10, we use the CelebA setup provided (the only 3-channel color dataset provided) and thus use $\beta = 3.0$, latent space size of 25 and 30 epochs instead. Note that we report "On-Learned-Manifold Adversarial Training" from (Stutz et al., 2019) which uses class-specific VAE-GANs. That is, 10 separate VAE-GAN architectures are trained for both FashionMNIST and CIFAR-10 datasets.

**Discussion on Hyperparameters of Generative Models.** As normalizing flows directly optimize log-likelihood of the data, there are no hyperparameters in their loss function. Additionally, the normalizing flow models that we use have a fixed latent dimension equal to the input dimension due to their architectural design. This is in contrast to VAE-GAN used in (Stutz et al., 2019) where the training involves optimizing separate losses for three networks (namely, encoder, decoder and discriminator) concurrently. Coefficients called $\beta$, $\gamma$ and $\eta$ are used to scale different loss terms involved such as reconstruction, decoder and discriminator loss. Additionally, the latent size for VAE-GAN is hand-picked for each dataset.

**Classifiers.** For MNIST, we use the Adam optimizer with a learning rate of 0.001 and weight decay of 0.001. We train LeNet-5 classifiers for 20 epochs with exponential learning decay of rate 0.1 for 10.000 steps. For FashionMNIST, we use the training setup used in (Stutz et al., 2019). That is, we use Adam optimizer with a learning rate of 0.01 and weight decay of 0.0001. We train classifiers for 20 epochs with exponential learning decay of rate 0.9 for 500 steps. For CIFAR10/100, we use the training setup used in (DeVries & Taylor, 2017; Zhang et al., 2018a). More precisely, we use Stochastic Gradient Descent (SGD) with a batch size of 128, a learning rate of 0.1, weight decay of 0.0005, and Nesterov momentum of 0.9. We train ResNet-18 classifiers for 200 epochs and multiply the learning rate by 0.2 at epochs $\{60, 120, 160\}$. For SVHN, we use the same optimizer with a weight decay of 0.0001. We train ResNet-18 classifiers for 120 epochs and multiply the learning rate by 0.1 at epochs $\{30, 60, 90\}$.

**Data Augmentation.** For CIFAR-10/100, we use standard data augmentation akin to (Zagoruyko & Komodakis, 2016). That is, we zero-pad images with 4 pixels on each side, take a random crop of size $32 \times 32$, and then mirror the resulting image horizontally with $50\%$ probability. We use such data augmentation for both training the generative and the classifier models. Hence, our normalizing flow model is capable of encoding-decoding operations on augmented samples as well. Advanced data augmentation baselines we use in Table 1 (DeVries & Taylor, 2017; Zhang et al., 2018a) also include the same standard data augmentations. However, (Stutz et al., 2019) does not use data augmentation in their generative model. To provide a more direct comparison between the performance of two generative models, in §B.3 we conduct an additional study without any data augmentations.

# B. Additional Results

## B.1. Results on MNIST

Table 5 summarizes our results on MNIST in full data regime. Although the baseline has very good performances on this dataset, we observe improved generalization.

*Table 5. Train and test accuracy* (%) as well as *loss* on **MNIST**. Comparison with standard training, versus ours—latent-space perturbations ($\mathcal{P}_{rand}$ & $\mathcal{P}_{adv}$).

| Perturbation | Train Accuracy | Train Loss | Test Accuracy | Test Loss |
|---|---|---|---|---|
| Standard | 99.80 | 0.0069 | 99.24 | 0.0288 |
| $\mathcal{P}_{rand}^{\ell_\infty}, \epsilon=0.15$ | 99.78 | 0.0076 | 99.28 | 0.0262 |
| $\mathcal{P}_{adv}^{\ell_\infty}, \epsilon=0.05, \alpha=0.01, k=10$ | 99.26 | 0.0230 | 99.43 | 0.0216 |

## B.2. Results on SVHN

Table 6 summarizes our results on SVHN in low-data regime. Similarly to §3.2, we conduct a cross-dataset experiment between CIFAR-10 and SVHN, where we use pre-trained normalizing flows on CIFAR-10 to train classifiers on SVHN. To provide a baseline on the effect of using different datasets for normalizing flows and classifiers, we also provide results with pre-training on SVHN. Latent attacks transferred from CIFAR-10 achieve superior performance to direct pre-training on SVHN, indicating that transferring augmentations across datasets is promising.

*Table 6. Test accuracy* (%) on **SVHN**, in the *low-data regime*, where we use 5% of the training set and the full test set. Comparison with normalizing flows trained on **CIFAR-10**, versus **SVHN**.

| | SVHN | | | CIFAR-10 | |
|---|---|---|---|---|---|
| Perturbation | Test | Improvement | Perturbation | Test | Improvement |
| Standard | 81.2 | – | Standard | 81.2 | – |
| $\mathcal{P}_{rand}^{\ell_2}, \epsilon=15.0$ | 84.9 | +3.7 | $\mathcal{P}_{rand}^{\ell_2}, \epsilon=15.0$ | 90.0 | +8.8 |
| $\mathcal{P}_{adv}^{\ell_2}, \epsilon=0.5, \alpha=0.25, k=2$ | 86.9 | **+5.7** | $\mathcal{P}_{adv}^{\ell_2}, \epsilon=0.3, \alpha=0.15, k=2$ | 90.5 | **+9.3** |

## B.3. Additional Results on CIFAR-10

**Results without Data Augmentation.** To provide a direct comparison between two generative models and eliminate the effect of data augmentation, we run additional experiments. Table 7 shows results for our latent perturbations without any data augmentation to train the normalizing flow and the classifier. In line with our FashionMNIST results in §3.3, we observe that both Randomized–LA and Adversarial–LA overperform standard baseline and VAE-GAN based approach.

*Table 7. Test accuracy* (%) on **CIFAR-10**, in the *low-data regime*, where we use 5% of the training set and the full test set, without data augmentation.

| Method | Low-data |
|---|---|
| Standard | 49.8 |
| VAE-GAN (Stutz et al., 2019) | 49.4 |
| Randomized–LA (ours) | 54.9 |
| Adversarial–LA (ours) | 58.2 |

**Results with Different Attack Parameters.** In Table 8, we provide results with varying hyperparameters for the different attacks. Observe that in the "higher" Adversarial–LA perturbation setting–where $\epsilon = 2.0$, the classifier still didn't fully fit to the training set but the test performance is above the standard baseline.

**Multi-step Training.** We run additional experiments where we sequentially apply different attack hyperparameters in a multi-step training with weaker perturbations to increase the performance on the test set. The results are listed in Table 8, denoted with +.

*Table 8. Train and test accuracy* (%) as well as *loss* on **CIFAR-10**, using ResNet-18. All of the models are trained with the same hyperparameters listed in §A.2. Perturbations listed with the + sign indicates a multi-step training. For example, last row lists the result of the model trained with $P_{adv}^{\ell_2}, \epsilon = 2.0, \alpha = 1.5, k = 2$ for 130 epochs, $P_{rand}, \epsilon = 0.25$ for 40 epochs and $P_{rand}^{\ell_2}, \epsilon = 10.0$ for 30 epochs. Note that, regardless of multi-step training, the hyperparameters, including the total number of training epochs (= 200), remain fixed across the experiments.

| Perturbation | Train Accuracy | Train Loss | Test Accuracy | Test Loss |
|---|---|---|---|---|
| *Baselines:* | | | | |
| Standard | 100.0 | 0.002 | 95.2 | 0.194 |
| $P_{PGD}^{\ell_2}, \epsilon=2.0, \alpha=0.5, k=10$ | 61.13 | 0.895 | 75.7 | 0.731 |
| $P_{PGD}^{\ell_\infty}, \epsilon=0.03, \alpha=0.008, k=10$ | 77.3 | 0.521 | 86.3 | 0.442 |
| *Ours:* | | | | |
| $\mathcal{P}_{rand}^{\ell_2}, \epsilon=10.0$ | 99.8 | 0.007 | 95.8 | 0.161 |
| $\mathcal{P}_{rand}^{\ell_\infty}, \epsilon=0.25$ | 99.5 | 0.015 | **96.3** | 0.142 |
| $+\mathcal{P}_{rand}, \epsilon=0.15$ | 100.0 | 0.002 | **96.4** | 0.133 |
| $\mathcal{P}_{adv}^{\ell_2}, \epsilon=1.0, \alpha=0.5, k=3$ | 99.9 | 0.005 | **96.6** | 0.126 |
| $\mathcal{P}_{adv}^{\ell_2}, \epsilon=2.0, \alpha=1.5, k=2$ | 89.1 | 0.214 | 95.8 | 0.134 |
| $+\mathcal{P}_{adv}^{\ell_2}, \epsilon=1.0, \alpha=0.75, k=2$ | 99.2 | 0.030 | 96.5 | 0.114 |
| $+\mathcal{P}_{adv}^{\ell_2}, \epsilon=0.75, \alpha=0.5, k=2$ | 99.7 | 0.011 | **96.7** | 0.115 |
| $+\mathcal{P}_{rand}, \epsilon=0.25$ | 100.0 | 0.002 | 96.5 | 0.132 |
| $+\mathcal{P}_{rand}^{\ell_2}, \epsilon=10.0$ | 100.0 | 0.002 | 96.6 | 0.131 |

**Generated Images.** Figure 3 depicts samples obtained with our Randomized-LA and Adversarial-LA methods. In contrast to random image space perturbations, we observe that both Randomized-LA and Adversarial-LA yield perturbations dependent on the semantic content of the input image.

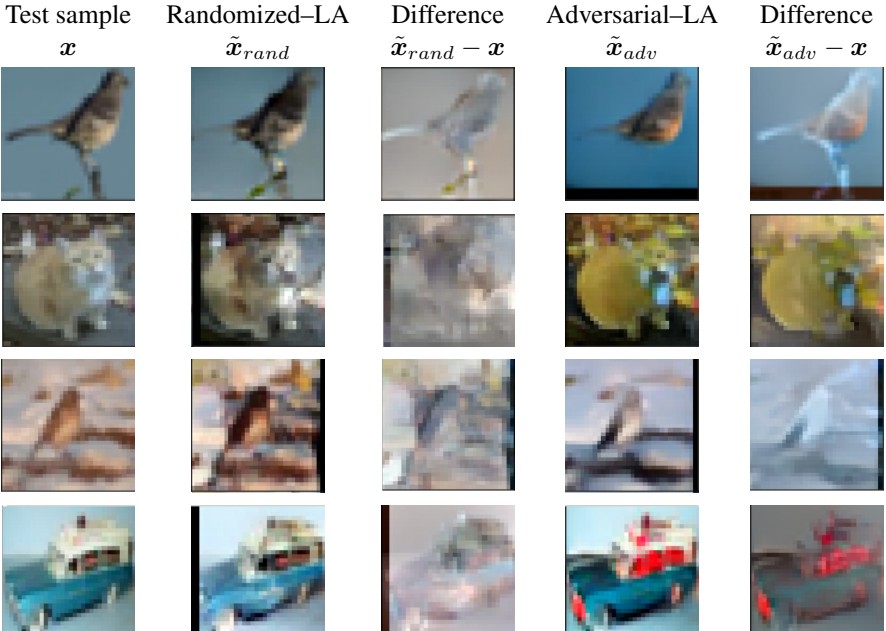

Test sample $\boldsymbol{x}$    Randomized–LA $\tilde{\boldsymbol{x}}_{rand}$    Difference $\tilde{\boldsymbol{x}}_{rand} - \boldsymbol{x}$    Adversarial–LA $\tilde{\boldsymbol{x}}_{adv}$    Difference $\tilde{\boldsymbol{x}}_{adv} - \boldsymbol{x}$

*Figure 3.* **Illustrative results of our latent space perturbations.** The first column depicts a randomly selected samples from the test set $\boldsymbol{x}$. We depict the outputs obtained with Eq. **Randomized–LA** and Eq. **Adversarial–LA** as well as their difference with the test sample $\boldsymbol{x}$. By observing the difference images, we see that the added perturbations depend on the semantic content of the input image.