# OpenReview forum: "Semantic Perturbations with Normalizing Flows for Improved Generalization"
_ICML.cc/2021/Workshop/INNF — INNF+ 2021 poster_

### Official Review · Reviewer_zq1b · 2021-06-11

**Rating:** Accept
**Confidence:** 4

**Summary:**

Authors propose using NFs to produce adversarial examples as data augmentation(DA) by making perturbations in the latent space. They show that this leads to better generalisation performance on classification tasks compared to standard DA methods.

**Justification For Rating:**

- Overall I find the paper nicely written and like the simplicity of the idea. The idea of using label information to find better data augmentation is sensible. I find it interesting that the adversarial examples that are found by increasing the cross entropy (and hence lowering predictive accuracy) is actually helping generalisation. I suspect that going too far with finding the adversarial example (i.e. using high values of k and alpha) will harm performance, and tuning these hyperparameters seem important. Hence it would be useful to show 1) how the performance varies for different values of k & alpha (e.g. showing plots of generalisation performance by fixing one and varying the other, or better still on a grid of values for k & alpha) 2) how robust the choice of k & alpha is for different NF models. One potential limitation of the method is that k & alpha might have to be retuned for each NF model. Since GLOW is explored here, it would be useful to also showcase it for a sufficiently different NF, perhaps a continuous-time flow like residual flows. 3) visualisation of these adversarial examples by showing their images for the CIFAR datasets used in the paper.
- There appears to be a disparity between the hypothesis 'on-maniforld adversarial attacks can improve generalisation' and the approach described in section 2.2. There is nothing ensuring that the perturbed z* (perturbed from latent rep z of datapoint x) lies in a region of high probability of the NF, since the perturbed z is only guided by the cross-entropy loss, hence x*=F^{-1}(z*) may have escaped the data manifold. If k and alpha are small enough, the value of p(z*) (the gaussian base density at z*) will be close to p(z) hence reasonably high, but this doesn't guarantee that p(x*) is also large. To really test the manifold hypothesis, you may want to consider using the NF likelihood to ensure that p(x*) isn't too small, e.g. use it to tune k for a given alpha.
- As mentioned in the discussion, the proposed approach is compatible with standard DA mehtods. It would be informative to show results for combinations, to see whether they synergise in practice, or actually are best used on their own. I can think of 3 ways of making combinations: 1) take the union of adversarial-LA images and standard DA images 2) apply adversarial-LA to standard DA images 3) apply standard DA to adversarial-LA images.
- For the cross-dataset experiments, the CIFAR-10 & CIFAR-100 used have the same size, so the motivation outlined in 3.2 is not so fitting. It would be more interesting/relevant to use a small fraction of ImageNet32 for this.

minor: line 94 typo F^{-1}(P(F(z))) -> F^{-1}(P(F(x)))

---

### Official Review · Reviewer_s9Pv · 2021-06-11

**Rating:** Borderline Accept
**Confidence:** 3

**Summary:**

The paper studies the data augmentation technique, which uses an invertible generative model (namely, normalizing flow) in order to construct semantically meaningful augmentations. The paper builds upon priors work, where VAE-GAN model was used for the same purpose. The augmentation is performed via additive perturbation of the latent code. Authors compare random and adversarial perturbations. The former adds Gaussian noise, while the latter learns the perturbation to 'fool' the classifier. The resulting augmentation approach performs better than competing methods both on the whole dataset and in the low-data regime.

**Justification For Rating:**

Below I list some directions in which the paper might be improved:

- Authors claim ( sec 4., second paragraph) that exact coding is the reason for the improved generalisation. I am not sure, that this is the case: what if we take the VAE model with similar performance (e.g. [1])? Overall, I think that it would be beneficial to add a comparison of the generative models used. For now, it might be the case that VAE-GAN just has a worse quality samples compared to GLOW.
[1] NVAE: A Deep Hierarchical Variational Autoencoder (2021). Arash Vahdat, Jan Kautz

- The method requires additional computational costs, which include
1. training a generative model
2. encoding image into the latent space
3. (For adversarial LAs) training a perturbation
4. Decoding new latent code back to the data space
I believe it would be beneficial to discuss this computational overhead in the paper and note how it compares to other methods. I assume that the VAE-GAN method would have a similar problem. But for other methods, it is less clear.

- I did not fully understand an experimental setup in a Low-data regime. Is the generative model also trained on 5% of the dataset or on the whole training data?
In the former case, it would be interesting to see the quality of the resulting generative model.
In the latter, I am wondering, what is the reason for the bad performance of VAE-GAN (it is worse than simple DA). It would be great if the authors can elaborate on this point in the paper.


A small remark:
- Line 094, col 1: Should be F(x) instead of F(z) (as in line 105)

---

### Official Review · Reviewer_D9H5 · 2021-06-11

**Rating:** Accept
**Confidence:** 4

**Summary:**

**Summary of paper**

Motivated by the challenges of other data augmentation techniques including latent space augmentations and adversarial training, this paper proposes data augmentations based on the latent space of a normalizing flow, which have exact latent variable inference and sampling given latent variable.  The paper claims that these are "on-manifold" perturbations that can be helpful especially in low-data regimes.

**Summary of review**

The paper is within scope of the workshop and provides a solid analysis of using normalizing flows for data augmentation. While a few claims about "locality" may be misleading, the empirical results seem to show the promise of the augmentation method.


**Justification For Rating:**

**Strengths:**
- Propose new on-manifold data augmentation method for training classifiers using normalizing flows.
- Demonstrate that a pretrained image normalizing flow can be used to create data augmentations for related classifiers. (in this case a flow based on CIFAR-10 is used to generate data augmentations for CIFAR-100).
- Overall demonstrate superiority of flow-based augmentations over prior methods.

**Weaknesses:**
- It seems that a small perturbation in the latent space could yield a very large perturbation in the original space, i.e., it could change the identity or class of the image.  How do you control for this possible issue? A priori, this is not clear whereas in normal adversarial examples, the intuitive assumption is that small changes to the image will not change the class.

- More generally, have you checked that these latent perturbations produce something reasonable in the original image space?  Images of perturbations would help intuitions.  Also, are the same classes actually clustered in latent space?  It would be helpful and important to demonstrate either theoretically or empirically that images of the same class are near each other.

- Related to the above comments, the paper mentions that flows map "any local neighborhood of a latent code to a local neighborhood of the original sample." (L213).  I don't think this is true.  It does map it to some continuous region that includes the original sample but the "locality" may no longer be true as an invertible model does not in general have any bounds on the Lipschitz constants in either direction and can thus arbitrarily skew, stretch or warp the space.


**Other comments or questions**
- The paper lacks a discussion of the choice of normalizing flow and why this was chosen.  Not all normalizing flows are the same. Residual flows are quite different than coupling-based flows like RealNVP.  It would be great to compare a few major classes of flows to see which flow structures work and which may not.

- Can you compare the computational costs of different augmentation methods?  Training a GLOW normalizing flow is quite expensive and adversarial updates is also expensive.  At least some discussion about this would be helpful.

- Why sample from Gaussian and then clip?  Why not just use a boundary attack like in adversarial attacks?  You could just treat the Gaussian noise as a gradient vector in adversarial attacks and then use a 1-step projection operator to the max $\epsilon$ ball.  For example, if $y$ is the Gaussian noise, this would be $\epsilon sign(y)$ as in the FGSM $\ell_{\infty}$$ attack.

- The adversarial attack can be generalized using the standard PGD attack by finding the steepest descent direction (e.g., normalization for $\ell_2$ and sign for $\ell_{\infty}$.

---

### Decision · Program_Chairs · 2021-06-14

Accept (poster)